# Irradiation-Induced Dysbiosis: The Compounding Effect of High-Fat Diet on Metabolic and Immune Functions in Mice

**DOI:** 10.3390/ijms24065631

**Published:** 2023-03-15

**Authors:** Briana K. Clifford, Nadia M. L. Amorim, Nadeem O. Kaakoush, Lykke Boysen, Nicodemus Tedla, David Goldstein, Edna C. Hardeman, David Simar

**Affiliations:** 1School of Health Sciences, UNSW, Sydney, NSW 2052, Australia; 2School of Nursing, Midwifery and Social Work, University of Queensland, Brisbane, QLD 4072, Australia; 3UTS Centenary Centre for Inflammation, School of Life Sciences, University of Technology, Sydney, NSW 2050, Australia; 4School of Biomedical Sciences, UNSW, Sydney, NSW 2052, Australia; 5The Danish Environmental Protection Agency, Ministry of Environment of Denmark, 5000 Odense, Denmark; 6Prince of Wales Clinical School, UNSW, Sydney, NSW 2052, Australia; 7Prince of Wales Hospital, Randwick, NSW 2031, Australia

**Keywords:** dysbiosis, adipogenesis, insulin resistance, inflammation, gut microbiota, irradiation, high-fat diet

## Abstract

The negative impact of irradiation or diet on the metabolic and immune profiles of cancer survivors have been previously demonstrated. The gut microbiota plays a critical role in regulating these functions and is highly sensitive to cancer therapies. The aim of this study was to investigate the effect of irradiation and diet on the gut microbiota and metabolic or immune functions. We exposed C57Bl/6J mice to a single dose of 6 Gy radiation and after 5 weeks, fed them a chow or high-fat diet (HFD) for 12 weeks. We characterised their faecal microbiota, metabolic (whole body and adipose tissue) functions, and systemic (multiplex cytokine, chemokine assay, and immune cell profiling) and adipose tissue inflammatory profiles (immune cell profiling). At the end of the study, we observed a compounding effect of irradiation and diet on the metabolic and immune profiles of adipose tissue, with exposed mice fed a HFD displaying a greater inflammatory signature and impaired metabolism. Mice fed a HFD also showed altered microbiota, irrespective of irradiation status. An altered diet may exacerbate the detrimental effects of irradiation on both the metabolic and inflammatory profiles. This could have implications for the diagnosis and prevention of metabolic complications in cancer survivors exposed to radiation.

## 1. Introduction

With the progressive improvement in the diagnosis and treatment of childhood cancer, survival rates have now reached more than 80% [1,2]. However, more than 95% of survivors will develop chronic clinicopathological conditions later in life as a consequence of their cancer or the associated treatments [3]. Such conditions include obesity and insulin resistance, some of the most frequent metabolic complications in childhood cancer survivors, with cancer treatments identified as key risk factors [4,5]. We and others have shown that irradiation in particular, results in impaired metabolic profile and function in long-term childhood cancer survivors [4,6,7]. Such metabolic complications have been further linked to altered immune cell function and systemic inflammation [8,9].

There is now accumulating evidence that the gut microbiota plays a critical role in the regulation of immune and metabolic functions [10,11,12], and disturbance of the gut microbiota, or dysbiosis, can lead to both inflammatory and metabolic diseases [13,14,15]. Acute and chronic changes in the gut microbiota, as well as gastrointestinal symptoms, have been reported during the course of cancer treatment, particularly following exposure to radiation [16,17,18]. Such changes suggest that irradiation-induced dysbiosis could be linked to immune disturbances and metabolic complications in cancer survivors. Changes in diet can also influence the composition and function of the gut microbiota, as well as metabolic function, particularly in the context of metabolic diseases [14,15,19]. However, it remains unclear how irradiation and changes in diet, which are common during cancer treatment, can in combination affect the gut microbiota and metabolic function. In this study, we hypothesised that the combination of irradiation and high-fat diet (HFD) would result in dysbiosis, leading to systemic inflammation and metabolic dysfunction. To test this hypothesis, we developed a mouse model of irradiation (single 6 Gy exposure or sham treatment) and HFD (12 weeks of HFD initiated 5 weeks post-irradiation) and investigated changes in the gut microbiota and systemic immune profile, and characterised immune cells and metabolic functions in the adipose tissue following irradiation.

## 2. Results

### 2.1. Whole Body Metabolic Profile

Seventeen weeks post-irradiation (including 12 weeks of HFD), we found a significant effect of irradiation (*p* = 0.028) and HFD (*p* < 0.0001) on body weight, with irradiation decreasing body weight, while HFD had the opposite effect (Table 1). There was a significant effect of diet on body fat (*p* < 0.0001), with high-fat-fed mice showing higher body fat, whilst we found a trend towards an effect of irradiation on body fat (*p* = 0.053), with irradiated mice showing lower body fat. We also found a significant HFD effect on blood glucose (*p* < 0.0001), with mice fed a HFD exhibiting higher blood glucose levels than those fed a chow diet. Insulin levels remained unaffected in response to both treatments.

### 2.2. Effect of Irradiation and Diet on Systemic Inflammation and Immune Cell Profiles

Here we tested the difference in individual cytokines between the four treatment groups at 17 weeks post-irradiation. We identified a significant interaction effect for a number of pro- and anti-inflammatory cytokines (Figure 1). Post hoc analyses demonstrated a significant effect of irradiation on IL-10, IP-10/CXCL10, MCP-1/CCL2, and MIP-1β/CCL3, with irradiation leading to an elevation of these cytokines’ levels. We also found a HFD effect on IL-5, IL-9, IL-13, IP-10/CXCL10, MCP-1/CCL2, and IL-22, with higher concentrations observed in the HFD condition, except for anti-inflammatory cytokine IL-22, which was lower in the HFD condition. There was a compounding effect of HFD and irradiation on IL-10, IP-10/CXCL10, IL-13, GROα/CXCL1, MCP3/CCL7, Eotaxin/CCL11, and MIP-1β/CCL3 (Figure 1. Panels A–K), with the association of both treatments leading to an increase in these cytokines and chemokines. We did not find any effect of irradiation or high-fat diet on several cytokines, including IL-1β, IL-18, IL-6, IFN-γ, and TNF-α (Appendix A).

A PCoA was generated to assess the effect of irradiation and diet on the composite inflammatory profile of mice. We identified a clustering of a subset of irradiated mice from both chow and high-fat-fed groups (Figure 1H). Multiple correlation showed that the clustering of the irradiated mice was driven by higher concentrations of Eotaxin (CCL11), GROα (CXCL1), IL-9, and IL-18.

We used a distance-based multivariate linear model (DistLM) to assess body weight, fat mass, irradiation status, and diet as predictive variables on the mice inflammatory profile, and found a significant effect of irradiation (*p* = 0.017), but no effect of body weight (*p* = 0.83), body fat (*p* = 0.77), or diet (*p* = 0.45) on the inflammatory profile.

### 2.3. Innate and Adaptive Immune Cell Phenotypic Profiles

In the face of increased circulating cytokines and the potential impact of irradiation, we investigated the profile of immune cells contained in the spleen as a surrogate marker of circulating leukocytes (Figure 2). We found a significant interaction effect between irradiation and HFD on the frequency of T helper cells. Our post hoc analysis revealed a significant increase in T helper cells in chow-fed irradiated mice compared to chow-fed non-irradiated (control group) mice, and in high-fat-fed irradiated mice compared to high-fat-fed non-irradiated mice (Figure 2A), suggesting that irradiation could drive an increase in circulating T helper lymphocytes. We did not find any effect of irradiation or high-fat diet on any other immune cells investigated here (Figure 2).

### 2.4. Immune Activation in Adipose Tissue

Immune activation has been reported in response to irradiation or HFD in key organs involved in metabolic functions, including the adipose tissue and skeletal muscle, and inflammation plays a crucial role in the development of metabolic diseases. This led us to interrogate the profile of infiltrated immune cells in the adipose tissue. Using flow cytometry, we established the frequency of immune cells in the epididymal fat depots (Figure 3) and found that both irradiation and HFD increased the infiltration of haematopoietic cells (CD45+, Figure 3A). We further found that HFD significantly increased the proportion of T cytotoxic cells, while decreasing the proportion of natural killer cells (Figure 3C,E). HFD also increased the proportion of activated macrophages (F4/80+CD11b+; Figure 3G), as well as pro-inflammatory or M1 macrophages (F4/80+CD11b+CD11c+; Figure 3H). We further identified a significant interaction effect between irradiation and HFD on M1 macrophages. Post hoc analysis revealed higher M1 macrophages in non-irradiated high-fat-fed mice compared to non-irradiated chow-fed mice, in irradiated high-fat-fed mice compared to irradiated chow-fed mice, as well as in irradiated high-fat-fed compared to non-irradiated high-fat-fed mice, suggesting a compounding effect of irradiation on HFD to increase M1 macrophage infiltration (Figure 3H).

### 2.5. Adipose Tissue Metabolic Functions

To test the impact of the remodelling of the immune profile of adipose tissue on adipocyte functions, we assessed the differentiation potential of preadipocytes upon adipogenic induction using flow cytometry (Figure 4A). Adipogenic induction increased the heterogeneity of cell granularity (Figure 4A), with HFD reducing the proportion of highly differentiated cells (R4), suggestive of impaired adipogenesis (Figure 4B). Irradiation further increased the proportion of undifferentiated (R1) and poorly differentiated (R2) cells, also supporting a negative effect of irradiation on adipocyte differentiation (Figure 4B). We then quantified the proportion of differentiated adipocytes (R3+R4) and observed a lower proportion of differentiated adipocytes in high-fat fed mice (Figure 4C). We further observed an interaction effect between irradiation and high-fat-diet, with post hoc analysis supporting a reduced proportion of differentiated adipocytes in irradiated chow-fed mice compared to non-irradiated chow-fed mice (control group) and in non-irradiated high-fat-fed mice compared to the same control group. We then measured the level of insulin receptor on differentiated adipocytes as an indicator of adipogenesis and found that high-fat-fed mice showed lower levels of insulin receptor compared to chow-fed mice (Figure 4D). To further characterise the metabolic functions of adipocytes, we assessed the phosphorylation level of Akt on its Ser473 residues upon insulin stimulation (Figure 4E). Phosphorylation of Akt on Ser473 has been demonstrated to be a key element of insulin signalling, facilitating glucose transporter translocation to the plasma membrane, and supporting glucose uptake in adipocytes. Both irradiation and high-fat diet negatively affected insulin-stimulated Akt phosphorylation, suggestive of insulin resistance in adipocytes from irradiated mice and mice fed a high-fat diet. These results suggest that separately or in combination, irradiation and high-fat diet contribute to impaired adipogenesis and adipocyte metabolic functions, potentially contributing to whole-body metabolic complications.

### 2.6. Effect of Irradiation and Diet on Gut Microbiota Diversity and Profile

To investigate the contribution of irradiation and high-fat diet to dysbiosis as a potential mechanism driving the inflammatory profile, we characterised gut microbiota diversity in our four groups of mice. We found a significant HFD effect on the α-diversity, a measure of species richness (*p* < 0.0001, Figure 5A), with no effect of irradiation. We found no effect of diet, irradiation, or any interaction on species evenness (Figure 5B), or Shannon index (Figure 5C). We visualised β-diversity with a principal coordinates analysis (PCoA) on Bray–Curtis dissimilarities (Figure 5D) and found that HFD groups clustered separately to the groups fed a chow diet. The HFD cluster was driven by high relative abundance of *Bifidobacterium_*OTU001 and *Allobaculum_*OTU002. In contrast, the chow-fed cluster was characterised by enriched relative abundance of *Erysipelotrichaceae_*OTU003 and *Lachnospiraceae_*OTU005. Within the chow-fed cluster, abundance of *Erysipelotrichaceae_*OTU003 was correlated with a clustering of irradiated mice. We used a DistLM to assess irradiation, diet, systemic markers of inflammation, and demographic characteristics as predictors of change in the gut microbiota. High-fat diet (Pseudo-F = 12.7, df = 22, *p* < 0.001) and fat mass (Pseudo-F = 9.1, df = 22, *p* < 0.001) were significant predictors of a difference in composition between groups. Irradiation and markers of inflammation did not predict a difference in microbiota composition between groups (*p* > 0.05).

## 3. Discussion

In this study, we investigated the effect of total-body irradiation and diet to elucidate their respective and compound impacts on the gut microbiota and immune and metabolic functions in mice. In line with our previous reports [6,20], we found that high-fat diet showed a stronger impact on body composition and glucose metabolism than irradiation. However, we also identified that these changes were linked to a substantial impact of high-fat diet on the composition of the gut microbiota, potentially masking the effect of irradiation alone. Surprisingly, despite its limited impact on the gut microbiota, irradiation led to more significant remodelling of the systemic profile of cytokines and immune cells. Irrespective of the magnitude of these immune changes, both high-fat diet and irradiation led to an increased proportion of pro-inflammatory macrophages in the visceral adipose tissue, providing a potential novel mechanism for the impaired adipogenesis and the development of insulin resistance we report here in response to both high-fat diet and irradiation (Figure 6).

Children diagnosed with cancer experience dramatic changes in their dietary habits during cancer treatment and well into survivorship [21,22,23]. Such dramatic changes independently or in combination with cancer treatment exert a major impact on the increased cardio-metabolic complications risk reported in childhood cancer survivors [5,24]. Using high-fat diet in an animal model of irradiation, we and others have previously reported the negative effect of high-fat diet, independently or in combination with irradiation, on body composition or whole-body metabolism [6,20,25]. Consistent with previous reports, here we found that high-fat diet led to body weight gain, increase in fat mass, and hyperglycaemia, whilst irradiation resulted in a decrease in fat mass with no significant effect on glucose metabolism. The mechanisms linking overnutrition to body weight and fat gain or impaired glucose metabolism have now been well established [26]. However, how irradiation might affect the metabolic profile of the host remains unclear. Previous reports from our team and others have identified a reduction in adipogenic potential, impaired proliferative capacity, or increased apoptosis as potential mechanisms contributing to decreased fat depots in response to irradiation [20,25,27]. We further reported a significant remodelling of both the epigenome and the transcriptome of adipocytes associated with impaired adipogenic potential and metabolic functions 40 weeks post-irradiation [20]. Here, our novel findings demonstrate impaired adipogenic potential as early as 17 weeks following a single exposure to 6 Gy radiation. These results suggest that irradiation could act as a predisposing factor for the development of metabolic dysfunction in individuals under dietary stress.

We found a limited effect of high-fat diet on the composite systemic cytokine profile, with irradiation exerting a significant effect on this profile, suggesting a persisting impact of irradiation on systemic inflammation irrespective of diet. Repeated doses of radiation, as is typical in the treatment of childhood and adult cancers alike, is likely to result in an increased inflammatory signature [28]. While no dietary effect was demonstrated on the composite inflammatory profile, we observed an effect of diet alone and irradiation alone on some systemic cytokines/chemokines. Notably, there seemed to be a compounding effect of diet and irradiation, with higher concentrations of traditionally inflammatory cytokines and lower concentrations of traditionally anti-inflammatory cytokines. A high-fat diet has previously been shown to negatively modulate the gut microbiome and to increase inflammatory signatures in young healthy adults in the absence of irradiation [29]. Systemic inflammation and gut microbiota changes induced by irradiation in combination with a high-fat or western diet may represent a triggering factor for the long-term metabolic disturbance experienced by cancer survivors.

Although we found limited disturbances in the systemic profile of immune cells in response to high-fat diet in particular, immune features of the epididymal fat depots in response to both treatments were suggestive of active inflammation. A remodelling of the immune profile of adipose tissues has been reported in the context of obesity and is thought to play a critical role in the development of metabolic dysfunction [30,31]. Such changes have been suggested to be potentially driven by increased absorption of antigen in the gut that could further lead to the activation of T cells and their trafficking to the adipose tissue [32]. An increased frequency of infiltrated T cytotoxic cells in the visceral adipose tissue is thought to act as the initial event, further leading to the recruitment of activated macrophages and M1 macrophages in response to high-fat diet, eventually resulting in impaired adipogenesis and insulin resistance [30]. Consistent with this report, here we identified an increased frequency of T cytotoxic cells, activated macrophages, and M1 macrophages in the epididymal fat depots of high-fat-fed mice. Although irradiation did not result in immune changes of the same magnitude as high-fat diet, irradiated mice still showed an increase in infiltrated M1 macrophages in the adipose tissue. In a similar model of irradiation, we previously failed to identify changes in the overall expression of *F4/80* in stroma vascular cultures, suggesting the absence of changes in macrophage representation in those cultures [20]. In the present study, by using flow cytometry to specifically characterise activated macrophages, as well as M1 and M2 macrophages, we were able to identify subtle changes affecting the immune profile of adipose tissue, identifying a novel long-term impact of irradiation on the immune profile of adipose tissue.

Irrespective of its aetiology, adipose tissue immune activation and inflammation have been linked to impaired metabolic functions in a variety of animal models of obesity [30,31]. Here we found that changes in the adipose tissue immune profile towards a proinflammatory profile in response to both high-fat diet and irradiation were associated with impaired insulin-stimulated Akt phosphorylation, a critical component of the insulin signalling cascade [33]. These observations are consistent with our previous findings in muscle satellite cells and preadipocytes [20]. Inflammation has been reported to play a crucial role in the development of insulin resistance and to interfere with adipocyte differentiation [32,34]. Innate immune cells, including M1 macrophages, represent a critical source of pro-inflammatory cytokines, particularly IL-1β and IL-18, and their processing by the inflammasome has been shown to control both adipocyte differentiation and insulin resistance in adipose tissue [34]. Thus, in the face of an increased infiltration of M1 macrophages to the adipose tissue (and T cytotoxic cells for high-fat-fed mice), we report both impaired differentiation potential and impaired insulin signalling in adipocytes. Our findings suggest that both high-fat diet and irradiation result in adipose tissue inflammation and impaired metabolic function, potentially predisposing the host to whole-body metabolic dysfunction [33].

High-fat diet is a potent modulator of the gut microbiota [35,36,37], as confirmed by our present findings. While the impact of irradiation on the gut microbiota was more discrete compared to high-fat diet, irradiated chow-fed mice showed enriched relative abundance of Firmicutes *Erysipelotrichaceae.* Some species of *Erysipelotrichaceae* are highly immunogenic and have been strongly associated with TNFα in humans infected with HIV [38]. The abundance of this bacterial family has also been convincingly linked to host lipid metabolism and metabolic dysfunction [39], suggesting that irradiation in the absence of a high-fat diet may influence *Erysipelotrichaceae* abundance and in turn metabolic function. Our analysis showed that the microbiota composition in mice fed a HFD was driven by an enriched abundance of Firmicutes *Allobaculum*, in agreement with previous reports [40].

Although we present here important and novel information, our results must be considered with caution due to the limitations of our study. We found limited changes in the profile of systemic cytokines in response to both interventions. Increasing the number of mice included in the study could potentially allow the identification of more subtle changes in cytokine levels whilst increasing the statistical power of the study. We could not directly test the association between the increased systemic levels of cytokines and immune cells, or the increased frequency of infiltrated leukocytes with markers of adipogenesis and insulin sensitivity, as the limited amount of biological samples that could be collected from each mouse did not allow us to perform all measurements on the same mice. Thus, we could only correlate changes in cytokine levels with the changes affecting the gut microbiota. Similarly, we prioritised the detection of a variety of immune cells over the more precise characterisation of T helper subsets, which could have informed the potential source of systemic and adipose tissue inflammation. Irrespective of these limitations, the present study identifies potential novel mechanisms linking changes in the gut microbiota with metabolic complications in response to irradiation or high-fat diet.

In the present study, we report that high-fat diet and, to a lesser degree, irradiation, lead to changes in the gut microbiota. The resulting dysbiosis could be associated with a remodelling of the adipose tissue immune profile toward a pro-inflammatory profile, leading to impaired adipogenic functions and metabolic dysfunction. Our findings suggest that cancer survivors exposed to radiation and under nutritional stress should be carefully monitored for the development of glucose intolerance and insulin resistance, even in the absence of obvious signs of obesity. Carefully monitoring their diet following exposure to radiation could provide valuable information regarding their risk of developing metabolic complications. In addition, nutritional interventions aimed at normalising their gut microbiota could provide novel therapeutic strategies [41]. Future research should characterise the changes affecting the gut microbiota or the immune system of cancer survivors shortly after termination of their treatment to inform potential preventive or therapeutic interventions targeting metabolic complications.

## 4. Materials and Methods

### 4.1. Procedure

Ethical approval was granted by the Animal Research Ethics Committee at UNSW Sydney (project 14/47B) and procedures were performed in compliance with the NSW Animal Research Act 1985 and the National Health and Medical Research Council (Australian Code for the Care and Use of Animals for Scientific Purposes, 2013). This study design has been reported elsewhere in detail [6]. Briefly, four-week-old C57BL/6J male mice were obtained from Animal Resources Centre (Perth, WA, Australia). They were non-tumour-bearing mice, housed four to a cage at the standard 12:12 light/dark cycle, and fed a regular chow diet with food and water available ad libitum. At 5 weeks of age, half of the mice were exposed to a single dose of radiation (6 Gy, X-RAD 320, Precision X-ray, Madison, CT, US) and the other half was sham-treated by being placed in the irradiator in the absence of radiation as previously described [6]. Five weeks post-irradiation, mice were divided into two diet groups, and maintained for 12 weeks on chow or HFD. Faecal samples were collected upon completion of the intervention (17 weeks post-irradiation), snap frozen in liquid nitrogen, then stored at −80 °C. Fasting blood samples were collected at 17 weeks post-irradiation after a 16 h fast and glucose was measured with a glucometer (Accu-Check Performa, Roche Diabetes Care Australia, North Ryde, NSW, Australia), while insulin levels were assessed by ELISA (Unltrasensitive Mouse Insulin ELISA, Crystal Chem, Elk Grove Village, IL, USA). Body weight and fat mass were measured by magnetic resonance imaging (MRI) at 17 weeks post-irradiation. At the end of the study, all mice were anaesthetised using Ketamine/Xylazine solution (100 and 10 mg/kg/body weight respectively) injected intra-peritoneally. Mice were then euthanised by heart puncture, before removal of the spleen and epididymal fat as previously described [6]. Plasma was obtained by centrifugation for measurement of cytokines.

### 4.2. Extraction of DNA from Stool

DNA was purified from stool samples using a validated DNA extraction kit (QIAamp Fast DNA Stool Mini Kit) for use with the QIAcube (Qiagen, Venlo, The Netherlands). InhibitEX buffer, Buffer AL, Buffer AW1, and Buffer AW2 were prepared prior to running the experiment. Approximately 200 mg of frozen stool was sectioned from each sample and placed in a 2 mL microcentrifuge tube. Samples were then lysed by adding 1 mL of InhibitEX Buffer and vortexed for 1 min or until thoroughly homogenised. Samples were centrifuged at 20,000× *g* for 1 min to pellet stool particles. Proteinase K (25 µL) was then transferred into a separate tube and 600 µL of sample supernatant was added to the proteinase K. The remaining procedures were automated using QIAcube. Six hundred µL of Buffer AL was added to samples, which were then vortexed for 15 s and incubated for 10 min at 70 °C. Six hundred µL of 100% ethanol was added to the lysate and 600 µL of solution was added to the QIAamp spin column. The spin column was centrifuged at 20,000× *g* for 1 min and then transferred to a new tube, discarding the filtrate. Five hundred µL of Buffer AW1 was added to the spin column and centrifuged for 1 min. The spin column was then transferred to a new tube and the filtrate was discarded. Five hundred µL of Buffer AW2 was then added to the spin column and centrifuged for 3 min to eliminate the possibility of Buffer AW2 carryover. The QIAamp spin column was transferred into a new tube and 200 µL of Buffer ATE was added directly to the QIAamp membrane. The membrane was incubated at room temperature for 1 min and centrifuged at 20,000× *g* for 1 min to elute DNA. DNA samples were transferred to a 96-well plate, covered, and protected from light, and then sent to the Ramaciotti Centre for Genomics (UNSW) for amplicon sequencing with Illumina MiSeq 2 × 250 bp chemistry. Prior to sequencing, PCR was used to amplify 16S rRNA with Kapa HiFi Hotstart ReadyMix. Samples were incubated at 95 °C for 3 min, followed by 25 cycles of 95 °C for 30 s, 55 °C for 30 s, 72 °C for 30 s and then 72 °C for 5 min. Earth microbiome primers (515F-806R) were used to target the V4 region of the 16S rRNA gene.

### 4.3. Cytokine Multiplex Assay

A Cytokine and Chemokine 26-Plex Mouse ProcartaPlex^TM^ Panel 1 (Thermofischer Scientific, Waltham, MA, USA) was used to assess cytokine levels in mouse plasma samples, according to the manufacturer’s instructions. We used a Luminex MAGPIX instrument (Luminex Corporation, Northbrook, IL, USA) calibrated with MAGPIX Calibration and Performance Verification Kits (Millipore, Burlington, MA, USA) with xPONENT software (Luminex) to obtain data. Data were analysed using Multiplex Analyst software version 5.1 (Merck, Darmstadt, Germany) as the Median Fluorescent Intensity (MFI) using spline curve-fitting for calculating analyte concentrations in samples.

### 4.4. Spleen Immune Cell Profiling

After euthanasia, the spleen was quickly removed and placed on a 70 µm cell strainer, previously wetted using wash buffer (PBS, 1% BSA) and crushed into a 50 mL Falcon tube using a 10 mL syringe plunger. After washing the filter twice using the wash buffer, the cell suspension was centrifuged (5 min, 300× *g*) and the pellet was resuspended in 1× lysing solution (10×: 0.037% EDTA, 8.89% NH4Cl, 1% KHCO_3_) and incubated in the dark for 10 min. The cell suspension was washed once by centrifugation and the pellet resuspended in wash buffer before adjusting the cell concentration to 1 × 10^6^ per 100 µL. Cells were then incubated with Live Dead cell marker before staining with the relevant antibodies (CD45 V500, CD3 APC-Cy7, CD8a PE-CF594, CD4 BV421, NKp46 BV421, B220 PE-CF594, F4/80 PE, and Siglec-F BV421, all from BD Biosciences, San Jose, CA, USA except for F4/80 from eBioscience, ThermoFisher Scientific, Waltham, MA, USA). After a final wash, FACS analysis was performed on BD LSRFortessa™ flow cytometer using the Diva software (BD Biosciences), and analysed using Flow Jo software (Tree Star, Ashland, OR, USA). Leukocytes were gated based on side scatter (cell granularity) and forward scatter (cell size), after exclusion of dead cells and doublets, and further gated based on CD45. T cells were considered CD3+ and B220-/F4/80-, with T helper cells CD3+CD4+ and T cytotoxic cells as CD3+CD8+. CD3- cells were divided between B220+ (B cells) and NKp46+ (NK cells). F4/80+ cells were gated for SiglecF- (macrophages and monocytes) and SiglecF+ (eosinophils). The respective frequency was established for each population out of the total pool of CD45+ cells.

### 4.5. Adipose Tissue Metabolic Functions and Immune Cells Infiltration

Pre-adipocytes were obtained from the epididymal fat after dissection and adipose tissue dissociation using the “Adipose Tissue Dissociation Kit, Mouse and Rat” (Miltenyi Biotec, Bergisch Gladbach, Germany). Adipose tissue was digested in Enzyme Mix for 41 min at 37 °C on a gentleMACS™ Dissociator (Miltenyi Biotec) and the resulting cell suspension was filtered through a 70 μL cell strainer. After centrifugation, the supernatant was discarded to remove the mature adipocytes contained in the stroma vascular fraction. The cell pellet was dissolved in lysing solution to remove red blood cells, before neutralisation by addition of growth media (10% DMEM, 1% Penicillin/Streptomycin, Gibco, Billings, MT, USA). After centrifugation, the pellet was resuspended in growth media and the cell suspension was plated in 24-well plates kept in a 37 °C incubator. Cells were kept an optimal density by passaging when they reached 60–70% confluence [20].

Differentiation of precursor adipocytes was initiated at full confluence as previously described [20], before assessing metabolic functions in mature adipocytes. At the end of differentiation (Day 8), cells were trypsinised, centrifuged, and resuspended in DMEM 1% Fetal Bovine Serum (Gibco). The cell suspension was distributed in two 15 mL falcon tubes and stimulated with 0 μM or 1 μM Insulin for 5 min. Cells were then fixed in 16% paraformaldehyde, before permeabilisation by adding 100% methanol, at −20 °C for 30 min. After 3 washes, cells were stained using antibody against pAkt Ser473 AF488 to assess insulin signalling and CD220 APC (insulin receptor). After one wash, cells were immediately processed on a BD FACS Calibur Flow Cytometer and the data were analysed using Flow Jo software (Tree Star, Ashland, OR, USA) as previously described [42]. Adipogenesis was assessed by gating adipocytes based on forward scatter (cell size) and side scatter (cell granularity), with increased cell granularity representing increased lipid accumulation. Cells were gated in four equal regions based on side scatter (R1 to R4, from low to increased granularity) before analysis for pAkt Ser473 and CD220.

Infiltration of immune cells in adipose tissue was assessed on the stroma vascular fraction from the epididymal fat as previously described with slight adaptations ([43] and see above Spleen Immune Cell Profiling). In addition to the antibodies described above, we further added CD11b APC and CD11c BV786 (both from BD Biosciences), as well as CD206 AF488 (R&D Systems, Minneapolis, MN, USA). After a final wash, FACS analysis was performed on a BD LSRFortessa™ flow cytometer using the Diva software (BD Biosciences), and analysed using Flow Jo software (Tree Star, Ashland, OR, USA). The same gating strategy as described under Spleen Immune Cell Profiling was used with the addition of F4/80+ cells gated for CD11b+ (activated macrophages), F4/80+CD11b+CD11c+ (M1 macrophages), F4/80+CD11b+CD206+ (M2 macrophages) and SiglecF+ (eosinophils, Appendix A). The respective frequency was established for each population out of the total pool of infiltrated cells.

### 4.6. Raw Data and Statistical Analysis

Data analysis was performed using GraphPad Prism (Version 9, Boston, MA, USA), and PRIMER-e (PERMANOVA) (Version 6.1.13, Albany, Auckland, New Zealand). In GraphPad Prism, the normality of the distribution was tested using the Skewness and Kurtosis tests. A two-way Analysis of variance or ANOVA (or Kruskal Wallis test for non-parametric data) was performed to test the effect of irradiation and diet on alpha and beta microbiota diversity outcomes, and for individual cytokine analysis. When a significant interaction effect was found, post hoc analysis was completed with a Tukey correction. Data were presented as mean +/− standard deviation with the significance level set at *p* < 0.05.

Forward and reverse 16S rRNA gene reads were assembled and analysed using the mothur package (Version 1.39.1) as described in the MiSeq standard operating procedures [44,45]. There were, on average, 11,240 reads per sample. The data underwent quality filtering, chimera check, and alignment with the SILVA reference database, and then sequences were clustered using the average neighbour method, at a cut-off of 0.03, into operational taxonomic units (OTUs) employing the RDP reference database (trainset 16_022016). The OTUs represent the approximate taxonomic level of species. Microbiota analysis included analysis of α-diversity measures such as richness, evenness and Shannon index, and β-diversity measures (Principal Coordinates Analysis) against predictors such as time and treatment. All statistical analyses were performed using PRIMER-e (PERMANOVA) v6.1.13.

## Figures and Tables

**Figure 1 ijms-24-05631-f001:**
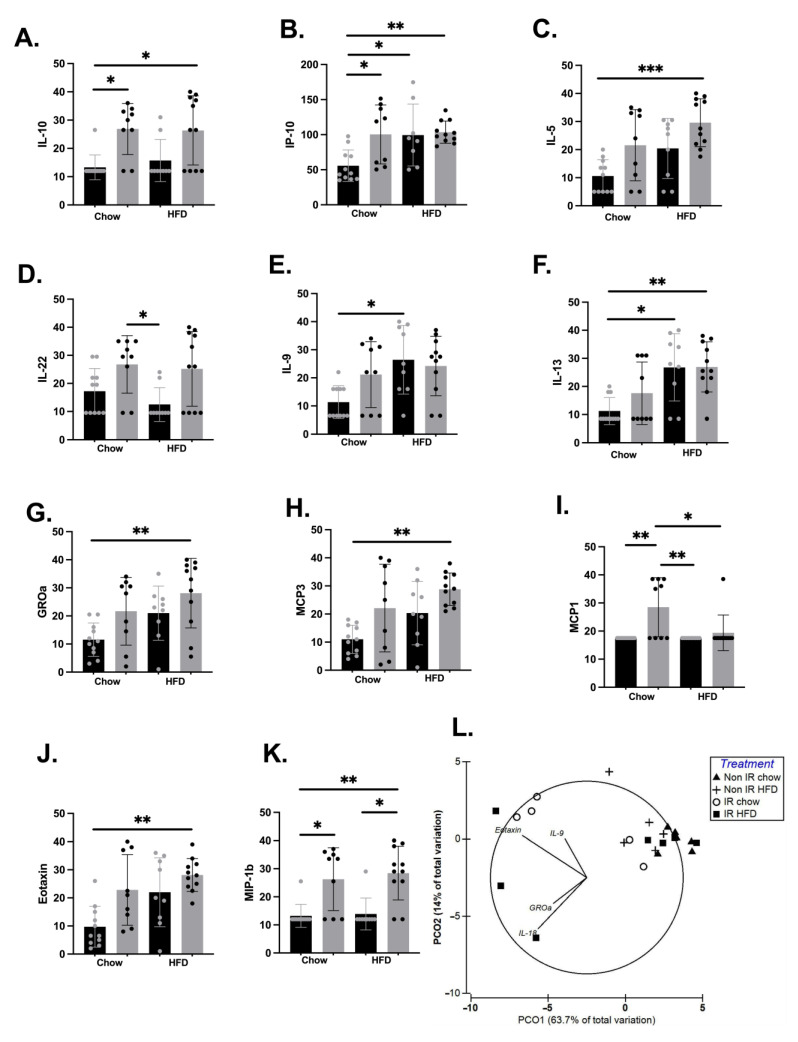
Inflammatory markers with significant differences between groups. (**A**). Interleukin (IL)-10. (**B**). Interferon gamma-induced protein (IP)-10. (**C**). IL-5. (**D**). IL-22. (**E**). IL-9. (**F**). IL-13. (**G**). Growth Regulated Oncogene alpha (GROa). (**H**). Monocyte Chemoattractant Protein (MCP)3. (**I**). MCP1. (**J**). Eotaxin. (**K**). Macrophage inflammatory protein (MIP)-1β. (**L**). Principal-coordinate analysis of a Euclidean distance resemblance matrix generated from square-root-transformed data of the 26 cytokines and chemokines. Overlayed vectors were generated with multiple correlation (>0.4). * = *p* < 0.05, ** = *p* < 0.01, *** = *p* < 0.001. Black bars and grey dots represent non-irradiated mice; grey bars with black dots represent irradiated mice. HFD—high-fat diet; IR—irradiated group.

**Figure 2 ijms-24-05631-f002:**
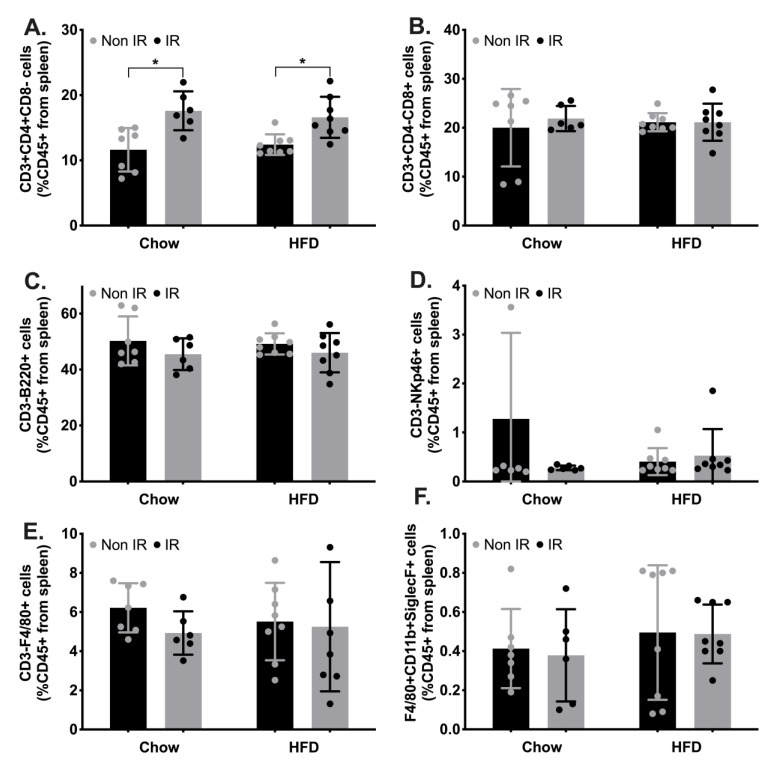
Immune cell profiling in the spleen. (**A**). T helper cells (CD45+CD3+CD4+CD8-). (**B**). T cytotoxic cells (CD45+CD3+CD4-CD8+). (**C**). B lymphocytes (CD45+CD3-B220+). (**D**). Natural killer cells (CD45+CD3-NKp46+). (**E**). Macrophages (CD45+F4/80+CD3-). (**F**). Eosinophils (CD45+F4/80+CD11b+SiglecF+). *—Interaction effect between irradiation (IRR) and high-fat diet (HFD) with significant difference between the two groups.

**Figure 3 ijms-24-05631-f003:**
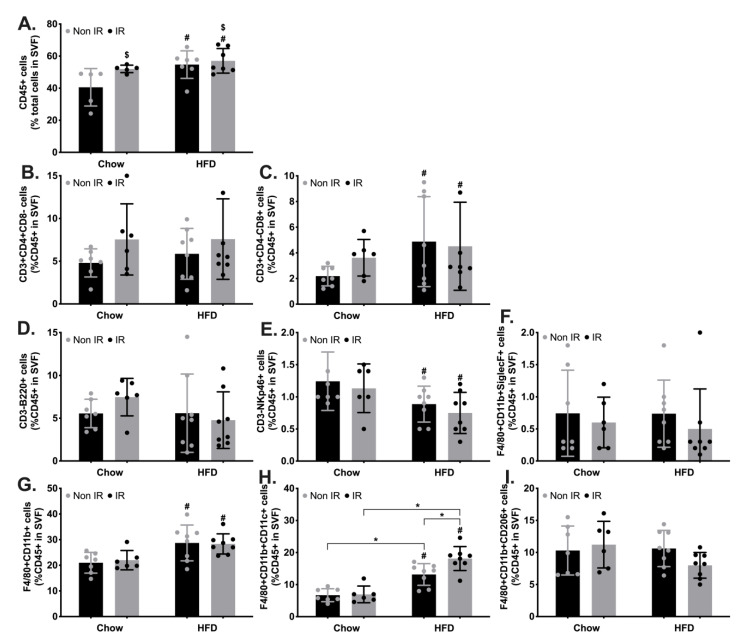
Immune cell profiling in epididymal fat tissue. (**A**). Haematopoietic (CD45+) cells. (**B**). T helper cells (CD45+CD3+CD4+CD8-). (**C**). T cytotoxic cells (CD45+CD3+CD4-CD8+). (**D**). B lymphocytes (CD45+CD3-B220+). (**E**). Natural killer cells (CD45+CD3-NKp46+). (**F**). Eosinophils (CD45+F4/80+CD11b+SiglecF+). (**G**). Activated macrophages (CD45+F4/80+CD11b+SiglecF-). (**H**). M1 macrophages (CD45+F4/80+CD11b+CD11c+). (**I**). (CD45+F4/80+CD11b+CD206+). $—irradiation effect; #—high-fat diet effect; *—interaction effect between irradiation (IRR) and high-fat diet (HFD), with significant difference between the two groups.

**Figure 4 ijms-24-05631-f004:**
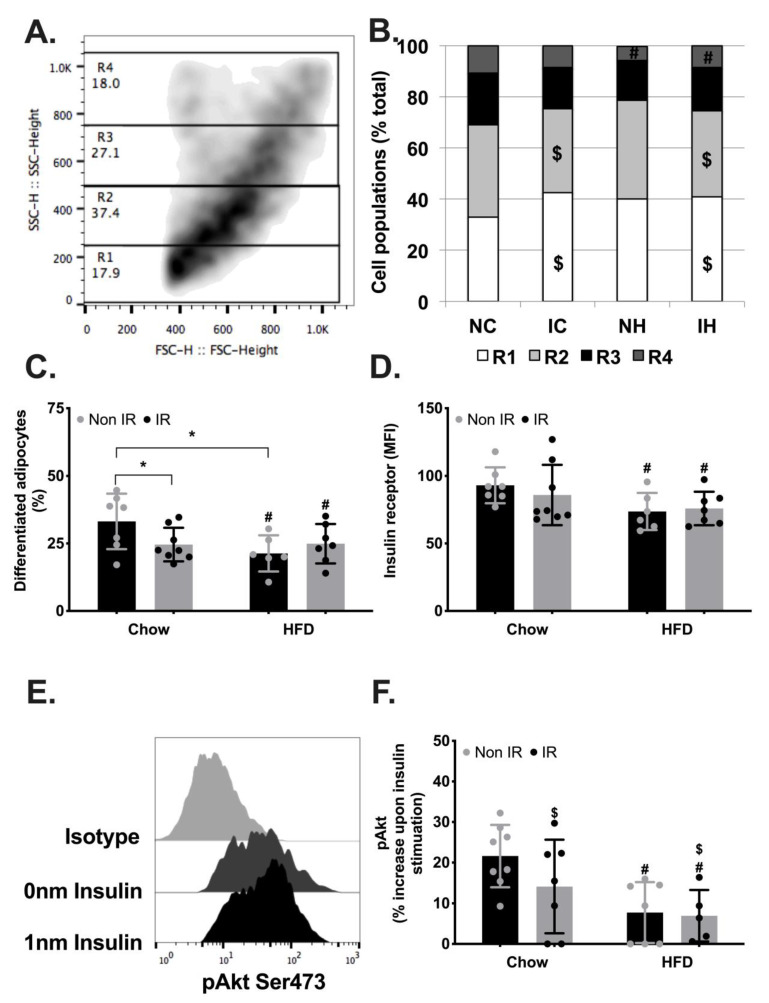
Adipocyte development and metabolic functions. (**A**). Representative flow cytometry analysis showing the four regions (R1-R4) defined by the granularity of the cells, indicative of their lipid accumulation at the end point of adipogenesis. (**B**). Quantification of the distribution of adipocytes across the four regions based on their differentiation (from not differentiated, R1, to highly differentiated, R4). (**C**). Proportion of highly differentiated adipocytes (contained in R3 and R4) at the end point of adipogenesis. (**D**). Expression of insulin receptor measured by flow cytometry on differentiated adipocytes. (**E**). Representative flow cytometry analysis of Akt phosphorylation on Ser473 residues in the absence (0 nM) or presence (1 nM) of insulin. (**F**). Percentage increase in Akt phosphorylation upon insulin stimulation. NC—non-irradiated chow diet, IC—irradiated (IR) chow diet, NH—non-irradiated high-fat diet (HFD), IH—irradiated high-fat diet. $—irradiation effect, #—high-fat diet effect, *—interaction effect between irradiation and high-fat diet with significant difference between the two groups.

**Figure 5 ijms-24-05631-f005:**
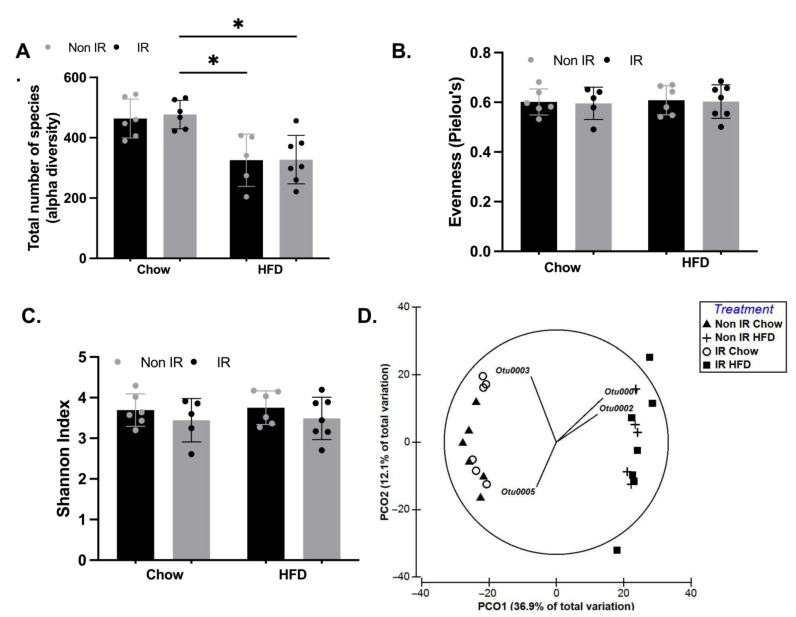
The effect of irradiation and diet on the gut microbiota diversity of mice. (**A**). Total number of OTUs; (**B**). Pielou’s evenness: intra-sample difference in relative abundance of OTUs; (**C**). Shannon index; (**D**). Principal-coordinate analysis (PCoA) of a Bray–Curtis resemblance matrix generated from square-root-transformed data with vectors generated by multiple correlations (>0.4). Non IR—Non-irradiated; IR—Irradiated; Chow = chow diet; HFD—High-fat diet. * = *p* < 0.05. In panels (**A**–**C**), black bars and grey dots represent non-irradiated mice; grey bars and black dots represent irradiated mice.

**Figure 6 ijms-24-05631-f006:**
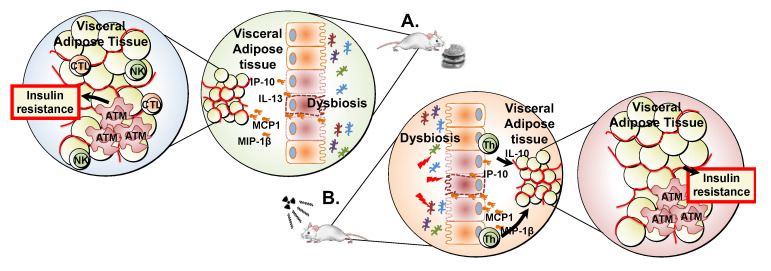
Effect of high-fat diet and irradiation on gut microbiota, immune profile, and metabolic functions. (**A**). High-fat diet leads to disturbances in the gut microbiota. Disturbance in gut microbiota could lead to intestinal inflammation and impaired gut permeability, resulting in systemic bacterial translocation. This in turn could be linked to systemic immune activation, remodelling of the systemic profile of cytokines, and increased infiltration of immune cells in the visceral adipose tissue. This change in the immune profile in adipose tissue could ultimately affect adipogenesis and result in insulin resistance [14]. (**B**). Irradiation leads to limited changes in the gut microbiota, but to a more dramatic remodelling of the systemic profile of cytokines and immune cells, potentially through direct immune activation [8]. The associated increased infiltration of proinflammatory macrophages to the visceral adipose tissue could ultimately impair adipogenesis and result in insulin resistance. CTL—cytotoxic T lymphocytes, NK—natural killer cells, ATM—adipose tissue macrophages, IP-10—Interferon gamma-induced protein-10, IL-13—Interleukin-13, MCP1—Monocyte Chemoattractant Protein-1, MIP-1β—Macrophage inflammatory protein-1 beta, IL-10—interleukin 10, Th—T helper cells.

**Table 1 ijms-24-05631-t001:** Metabolic data at 17 weeks post-irradiation.

	Chow Diet		High-Fat Diet				
	Non IR	IR	Non IR	IR	Irradiation (*p*-Value)	Diet (*p*-Value)	Interaction (*p*-Value)
Body weight (g)	29.8 ± 2.4	26.0 ± 1.3	43.4 ± 6.2	37.7 ± 5.4	0.028 *	<0.0001 *	0.616
Body Fat (g)	4.7 ± 0.8	3.8 ± 0.5	17.7 ± 4.0	13.8 ± 2.9	0.053	<0.0001 *	0.200
Blood Glucose (mM/L)	8.4 ± 1.9	9.4 ± 1.2	12.8 ± 1.5	13.1 ± 2.4	0.418	<0.0001 *	0.682
Insulin (µU/L)	10.5 ± 3.1	10.9 ± 4.4	16.0 ± 8.7	11.8 ± 4.2	0.459	0.214	0.355

All values expressed as mean ± standard deviation. Non-IR—Non-Irradiated Mice; IR—Irradiated Mice; Statistical significance * = *p* < 0.05; residual df = 16. Non IR Chow, *n* = 6; IR Chow, *n* = 4; Non IR High-Fat Diet, *n* = 5; IR High-Fat Diet, *n* = 5.

## Data Availability

Data can be made available by the corresponding author upon reasonable request.

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
