# Peer review of "Irradiation-Induced Dysbiosis: The Compounding Effect of High-Fat Diet on Metabolic and Immune Functions in Mice"

_ijms, 2023, doi:10.3390/ijms24065631_

Round 1

Reviewer 1 Report

General comments

In this mouse experimental study, the authors tested the hypothesis that a high-dose irradiation followed by a high-fat diet (HFD) induces dysbiosis, systemic inflammation along with adipose tissue metabolic dysfunction, by assessing gut microbiome, metabolic indicators, and immune profiles. HFD greatly affected body composition as well as glucose metabolism through changing the composition of the gut microbiota while a single-dose 6-Gy irradiation altered cytokine levels and immune cell profiles. Consequently, adipogenesis was impaired and insulin resistance was induced. The findings are considered important for assessing risks of chronic diseases such as diabetes mellitus and cardiovascular disease in childhood cancer survivors. The study design including animal procedures is appropriate, and the manuscript is well written. However, associations between impaired adipogenesis and immune/inflammatory profiles are not fully evaluated, which is a weakness of this study. The reviewer would like to know about whether radiation-induced increases in inflammatory cell and cytokine levels are statistically associated with alterations in adipocyte differentiation profiles and metabolic functions. Another weakness is the lack of T-helper subset data that might well support the authors’ argument that impact of irradiation on systemic inflammation, especially on circulating T-helper subsets (Figure 6), may be involved in impaired adipogenesis and insulin resistance following HFD in irradiated mice.

Specific comments

1.       Abstract: The dose “6 Gy” used for irradiation should be incorporated in the Method subsection. In the radiation health study field, effects of low-dose radiation exposures and dietary habits on risks of metabolic abnormalities in refugees from the Fukushima-Daiich nuclear accident are issues of current importance.

2.       Introduction: The authors would better to provide a notion on the procedure of this mouse study, i.e., 6-Gy exposure at 5 weeks of age and 12-week HFD (initiated at 5 weeks after irradiation), and its implication for assessments of metabolic dysregulation among childhood cancer survivors.

3.       Cytokine assessments: Some brief explanations about the results shown in Supplementary Material 1 are required in the text. Most cytokine levels shown in this figure did not significantly elevate in response to irradiation and/or HFD, which looks in part due to the lack of statistical power. Especially, somewhat elevated IL-1b and IL-18 levels are important for discussing about inflammasome activation due to irradiation.

4.       Interaction effect between irradiation and high fat diet: Are the interaction terms in Figures 2A and 4C negative while those in Figure 3H positive?

5.       Figure 6: More detailed explanations on the potential cellular and molecular mechanisms should be provided in the legend. Abbreviations, ATM, Th, and CTL should also be explained in the legend.

6.       Legend of Supplementary Material 1: Why not abbreviate Interleukins?

7.       References: Should be revised in accordance with the journal’s instruction.

8.       Overall, typos, spacing, and hyphenation should be carefully corrected.

Reviewer 2 Report

1.      The English writing of the manuscript needs improvement. Therefore, it could benefit greatly from professional editing to improve technical writing and English.

2.      Please mention your study limits and suggest some future research topics

3.      In References, the sources are written in different styles. Please update the reference list.  It is necessary to bring in accordance with the requirements of the journal for the design of References. If possible, indicate DOI.

4.      Please use some innovative keywords.

5.      Please mention your study limits in the abstract.

6.      The Conclusions should reflect what the practical application of the results obtained in this study is. In what climatic conditions should the recommendations of the authors be taken into account?

7.      The authors should increase their discussion on previous related research and highlight how their study is providing a different approach or adding significantly to what has been done. The authors have to explain what is the new here in comparison with the previous studies. The novelty of the current work should be highlighted in the introduction. Please try to mention a problem that needs solving - in other words, the research question underlying your study clearer.

Round 2

Reviewer 2 Report

 Accept in present form